# Intelligence Degradation in Long-Context LLMs:
# Critical Threshold Determination via Natural Length Distribution Analysis

## Abstract

Large Language Models (LLMs) exhibit a concerning phenomenon where performance catastrophically degrades when processing contexts approaching certain critical thresholds, even when the information remains relevant. This *intelligence degradation*—defined as a $>30\%$ drop in composite task performance—severely limits practical applications requiring long-context understanding.

This degradation exhibits a common underlying characteristic: models maintain strong performance up to a critical threshold, but once exceeded, performance collapses catastrophically. We term this phenomenon *shallow long-context adaptation*—models adapt their processing primarily for short to medium contexts but fail to maintain performance as length approaches critical thresholds.

This paper presents three main contributions: (1) **Natural Length Distribution Analysis Methodology**: We develop a rigorous experimental approach that uses each sample's natural token length (without truncation or padding) to provide stronger causal evidence that performance degradation results from context length itself rather than artifacts from text manipulation. (2) **Critical Threshold Determination**: Through comprehensive experiments on a mixed dataset (1,000 samples covering 5%-95% of context length), we precisely identify the critical threshold for Qwen2.5-7B at 40-50% of maximum context length, where F1 scores drop catastrophically from 0.55-0.56 to 0.3—a 45.5% performance degradation. Our approach employs a five-method cross-validation framework to ensure robust threshold identification. (3) **Unified Framework**: We consolidate the

notion of shallow long-context adaptation, demonstrating how it explains the observed degradation patterns and providing a foundation for future mitigation strategies.

This work provides the first systematic characterization of intelligence degradation in open-source Qwen models using natural length distribution analysis, offering practical guidance for deploying LLMs in long-context scenarios.

## 1. Introduction

Large Language Models (LLMs) have achieved remarkable success across diverse tasks, with context lengths expanding from 512 tokens to 128K or beyond. However, a critical vulnerability has emerged: when context length reaches certain critical thresholds, models exhibit catastrophic performance degradation—a phenomenon we term *intelligence degradation*. We define intelligence operationally as the model's performance on a specific task (e.g., reading comprehension) when using a given context length, quantified by the F1 score—see Theorem 3.1 for formal treatment. This degradation occurs even when all context information remains relevant and useful, suggesting fundamental limitations in how current architectures process long sequences.

**Motivating Example**. Consider a reading comprehension task where a model must answer questions based on a long narrative. In the stable region (0-40% of maximum context length, approximately up to 51,200 tokens for a 128K context model), the model achieves F1 scores around 0.55-0.58 (mean 0.565), demonstrating strong comprehension. However, when the same task is presented at 50% of maximum context length (approximately 64,000 tokens), performance catastrophically drops to F1 $\approx 0.302$—a 45.5% degradation from the stable region mean—even though the relevant information remains present and accessible. This abrupt collapse occurs over a narrow 10% range (from 40% to 50%), suggesting that the model's processing capabilities fundamentally break down beyond a critical threshold, rather than gradually degrading.

This degradation exhibits a common underlying characteristic: models maintain strong performance up to a critical

---

[1]Anonymous Institution, Anonymous City, Anonymous Region, Anonymous Country. Correspondence to: Anonymous Author <anon.email@domain.com>.

Preliminary work. Under review by the International Conference on Machine Learning (ICML). Do not distribute.

threshold, but once exceeded, performance collapses catastrophically. We term this phenomenon *shallow long-context adaptation*—models adapt their processing primarily for short to medium contexts but fail to maintain performance as length approaches critical thresholds. This shallow adaptation explains why simple context length increases can trigger dramatic failures and offers a unifying framework for understanding various degradation phenomena observed in long-context LLMs.

Recent studies have documented this phenomenon in closed-source models (Du et al., 2025; Huang et al., 2025; He et al., 2025), but systematic investigation of open-source models remains limited. The Qwen series, widely adopted in both academia and industry, lacks comprehensive analysis of its degradation behavior. More critically, while degradation has been observed (Liu et al., 2023; Deng et al., 2024), *precise determination of critical thresholds* using rigorous experimental methodologies remains underexplored. Understanding where and why degradation occurs is crucial for practical long-context applications.

This paper addresses these gaps through a systematic investigation with three main contributions:

1. **Natural Length Distribution Analysis Methodology**: We develop a novel experimental approach that uses each sample's natural token length without truncation or padding, providing stronger causal evidence that performance degradation results from context length itself rather than artifacts introduced by text manipulation. This methodology eliminates potential biases from artificial context length control and enables more rigorous threshold determination.

2. **Critical Threshold Determination via Multi-Method Cross-Validation**: We precisely identify the critical degradation threshold for Qwen2.5-7B at 40-50% of maximum context length (128K tokens), where F1 scores drop catastrophically from 0.55-0.56 to 0.3—a 45.5% performance degradation. Our approach employs a five-method cross-validation framework (gradient analysis, second derivative analysis, binned statistics, percentile threshold, and sliding window) to ensure robust threshold identification, with the final threshold determined as the median of all method estimates.

3. **Unified Framework for Shallow Long-Context Adaptation**: We consolidate the notion of shallow long-context adaptation, demonstrating how it explains the observed degradation patterns. This unified framework provides a foundation for understanding why degradation occurs and guides future mitigation strategies. We show that shallow adaptation results from

multiple factors including training data bias, optimization shortcuts, RoPE encoding extrapolation failures, and attention dispersion at longer lengths.

Our work provides the first systematic characterization of intelligence degradation in open-source Qwen models using natural length distribution analysis. The remainder of this paper is organized as follows: Section 2 reviews related work and positions our contributions; Section 3 characterizes shallow long-context adaptation; Section 4 presents our methodology; Section 5 provides theoretical analysis of shallow adaptation; Section 6 presents experimental results and critical threshold analysis; Section 7 concludes and discusses future directions. Detailed experimental setups, additional results, and ablation studies are provided in the Appendix.

## 2. Related Work

Early research focused on extending context length through architectural innovations (Beltagy et al., 2020; Zaheer et al., 2020; Dai et al., 2019), primarily addressing efficiency rather than performance degradation. Recent studies document degradation (13.9% to 85%) (Du et al., 2025; Huang et al., 2025) and the "Lost in the Middle" phenomenon (Liu et al., 2023), but most focus on closed-source models and use artificial truncation/padding, introducing methodological artifacts.

**Key Limitations**: (1) Methodological artifacts from truncation/padding; (2) Limited model coverage (closed-source); (3) Imprecise threshold determination (visual inspection); (4) Lack of unified framework. Our work addresses these through natural length analysis, systematic open-source analysis, multi-method cross-validation, and a unified framework. Here compares our work with prior studies:

*Table 1.* Comparison of Our Work with Prior Studies

| Aspect | Prior Work | Our Work |
|---|---|---|
| Methodology | Truncation/Padding | Natural Length |
| Model Coverage | Closed-source | Open-source (Qwen) |
| Threshold Detection | Single method | Multi-method (5) |
| Theoretical Framework | Descriptive | Unified (3 perspectives) |
| Statistical Rigor | Visual inspection | Quantitative validation |
| Dataset Design | Single dataset | Mixed dataset |

Detailed comparison with prior work is provided in Appendix A.

## 3. Characterizing Shallow Long-Context Adaptation

We characterize *shallow long-context adaptation* as models maintaining strong performance up to critical thresholds but

failing catastrophically beyond them. This unified framework explains why degradation occurs and provides a foundation for understanding various long-context performance issues. The experimental evidence presented in Section 6 (three distinct performance regions with abrupt transition) directly supports this characterization, while the theoretical analysis in Section 5 provides mechanistic explanations for why shallow adaptation occurs.

### 3.1. Mechanisms of Shallow Adaptation

Shallow adaptation occurs due to multiple interconnected factors:

1. **Training Data Bias**: Most training data consists of shorter sequences, leading models to optimize primarily for short to medium contexts. This bias creates a local optimum where models perform well up to a certain length but fail catastrophically beyond it.

2. **Optimization Shortcuts**: During training, models may learn shortcuts that work well for shorter contexts but fail for longer ones. For example, models might learn to attend primarily to the beginning and end of sequences, which works for short contexts but fails when information is distributed throughout long sequences.

3. **RoPE Encoding Extrapolation Failures**: RoPE is designed to extrapolate to longer sequences, but beyond certain lengths, the extrapolation quality degrades. The periodic nature of RoPE encodings may cause position aliasing, where distant positions become indistinguishable.

4. **Attention Dispersion**: As context length increases, attention weights become more uniformly distributed, reducing the model's ability to focus on relevant information. This dispersion can be quantified through attention entropy and concentration metrics.

These mechanisms interact to create the shallow adaptation phenomenon, where models maintain performance up to a critical threshold but fail catastrophically beyond it. Detailed theoretical analysis is provided in Section 5 and Appendix C.

### 3.2. Defining Model Intelligence

We formalize *model intelligence* and *intelligence degradation*:

**Definition 3.1** (Model Intelligence). We define **model intelligence** $I(L)$ at context length $L$ for a specific task as the model's performance on that task when using context of length $L$, quantified by the F1 score:

$$I(L) = \text{F1}(L) \qquad (1)$$

where $\text{F1}(L)$ is the F1 score achieved by the model on the given task when processing a context of length $L$. For example, for the reading comprehension task, $I(L)$ represents the F1 score obtained when answering questions based on a context of length $L$.

This definition provides a direct, task-specific measure of model capability. The F1 score captures the balance between precision and recall, making it suitable for evaluating model performance on tasks such as reading comprehension and summarization.

**Definition 3.2** (Intelligence Degradation). The **degradation rate** at context length $L$ is:

$$D(L) = \frac{I(L) - I(L + \Delta L)}{I(L)} \times 100\% \qquad (2)$$

where $\Delta L$ is the context length increment.

**Definition 3.3** (Cliff-like Intelligence Degradation). A model exhibits **cliff-like intelligence degradation** at critical point $L_c$ if:

1. **Abruptness**: $D(L_c) > \theta$ where $\theta = 30\%$ (performance drop exceeds 30%)

2. **Persistence**: $I(L > L_c) < 0.5 \cdot I(L < L_c)$ (sustained low performance beyond the critical point)

This contrasts with *gradual degradation* where $D(L) \approx$ constant $\ll \theta$ for all $L$.

**Justification**: The F1 score provides a balanced measure that captures both precision and recall, making it suitable for evaluating model performance on tasks such as reading comprehension. In this paper, we focus on the reading comprehension task, where $I(L)$ directly represents the F1 score achieved when processing contexts of length $L$. Detailed evaluation methodology is in Appendix F.

## 4. Methodology

### 4.1. Natural Length Distribution Analysis

Unlike prior work that artificially truncates or pads contexts to specific lengths, we employ **natural length distribution analysis**. For each sample $s_i$, we: (1) compute its natural token count $t_i$ using tiktoken, (2) calculate context ratio $r_i = t_i/T_{\max}$ where $T_{\max} = 131,072$ (128K tokens), (3) evaluate performance $p_i = M(s_i)$ without any text manipulation, and (4) record the pair $(r_i, p_i)$ for analysis.

This approach eliminates artifacts from truncation/padding and provides stronger causal evidence that performance

degradation results from context length itself rather than text manipulation. Traditional truncation may remove important information, confounding length effects with information loss, while padding may introduce artificial tokens that affect attention patterns. Algorithm details are in Appendix B.

### 4.2. Experimental Framework

We evaluate Qwen2.5-7B (7B parameters, 128K max context) on a **mixed dataset** combining:

- **500 SQuAD samples**: Short-context reading comprehension (average 1K tokens, covering 5%-10% of max context)

- **500 NarrativeQA samples**: Long-context reading comprehension (average 95K tokens after filtering, covering 20%-95% of max context)

The mixed dataset provides comprehensive coverage from 5% to 95% of the context length spectrum, eliminating sample distribution bias that would occur if using only a single dataset. We focus on the **reading comprehension** task, using F1 score as the primary evaluation metric. All experiments use fixed seeds (seed=42) and deterministic generation (temperature=0). Total: 1,000 test samples. Detailed setup and evaluation metrics in Appendix G.

### 4.3. Critical Threshold Detection

We employ a **five-method cross-validation approach** to detect critical degradation thresholds. This framework uses gradient analysis, second derivative analysis, binned statistics, percentile threshold, and sliding window analysis. While Methods 1, 2, 4, and 5 share a common three-stage filtering strategy (multi-peak detection, rise-in-range filtering, sustained decline verification), they differ in their theoretical foundations and application contexts: Method 1 focuses on gradient-based peak identification, Method 2 emphasizes second-derivative analysis for acceleration points, Method 4 uses percentile-based filtering, and Method 5 employs sliding-window smoothing. Method 3 (Binned Statistics) uses a fundamentally different approach based on bin-wise aggregation, providing complementary validation.

The detection process employs a three-stage strategy: (1) **Multi-peak detection** identifies all local performance peaks in the 30%-60% range using a sliding window approach (window size $w = 5$, minimum peak height $h_{\min} = 0.3$); (2) **Rise-in-range filtering** excludes peaks followed by recovery within 10% of the peak ratio; (3) **Sustained decline verification** confirms that performance continues to decline after the peak without significant rebound (rebound threshold: 85% of peak value). The final threshold is determined

as the median of all successful method estimates, ensuring robustness against method-specific biases. A critical threshold is identified when the performance drop exceeds 30%. The high consistency across methods (std dev 1.2%) despite different theoretical foundations validates that the shared filtering strategy captures robust patterns in the data. Detailed algorithms and parameter sensitivity analysis are in Appendix G.4.

## 5. Theoretical Analysis

We analyze shallow adaptation from three complementary perspectives to understand why the critical threshold appears at 40-50% of maximum context length.

### 5.1. Information-Theoretic Perspective

From an information-theoretic perspective, shallow adaptation can be understood as a decrease in effective information transmission capacity as context length increases. We formalize this as a **conceptual framework** that provides explanatory insights into the degradation mechanism:

**Definition 5.1** (Information Transmission Efficiency). The **information transmission efficiency** $\eta(L)$ at context length $L$ is defined as:

$$\eta(L) = \frac{I(X;Y|L)}{H(X|L)} \tag{3}$$

where $I(X;Y|L)$ is the mutual information between input $X$ and output $Y$ conditioned on length $L$, and $H(X|L)$ is the entropy of the input conditioned on length $L$. This metric provides a theoretical framework for understanding how information flow degrades with context length, though direct measurement requires access to model internals.

As $L$ approaches the critical threshold $L_c$, $\eta(L)$ conceptually decreases due to: (1) **Information Bottleneck**: Long contexts create a bottleneck where the model must compress more information into a fixed representation space, leading to information loss. (2) **Noise Accumulation**: Irrelevant information accumulates, reducing the signal-to-noise ratio. (3) **Attention Saturation**: The attention mechanism becomes saturated, unable to effectively allocate attention weights. This conceptual framework provides a mechanistic explanation for why performance degrades, consistent with the observed experimental patterns. Detailed derivations and lower bounds are in Appendix D.1.

### 5.2. Attention Mechanism Analysis

The attention mechanism is central to understanding shallow adaptation. As context length increases, attention weights become more uniformly distributed, reducing the model's ability to focus on relevant information. We formalize this

through a **conceptual framework** based on measurable attention metrics:

**Definition 5.2** (Attention Concentration). The **attention concentration** $C(A, L)$ at context length $L$ is defined as:

$$C(A, L) = \frac{1}{L} \sum_{i=1}^{L} \max_j A_{ij}(L) \qquad (4)$$

where $A_{ij}(L)$ is the attention weight from position $i$ to position $j$ at context length $L$. Higher concentration indicates more focused attention. This metric can be directly computed from attention weights when model internals are accessible.

As $L$ increases, $C(A, L)$ theoretically decays exponentially while attention entropy $H(A|L) = -\frac{1}{L} \sum_{i,j} A_{ij}(L) \log A_{ij}(L)$ increases, reducing focus on relevant information. The critical threshold occurs when $H(A|L)$ exceeds a threshold causing catastrophic degradation. This conceptual framework provides a mechanistic explanation for the observed performance degradation: the uniform distribution of attention weights at longer lengths directly corresponds to the model's reduced ability to focus on relevant information, consistent with the observed performance collapse. Detailed dynamics and the attention-performance relationship are in Appendix D.2.

### 5.3. RoPE Extrapolation Analysis

Rotary Position Embedding (RoPE) is designed to extrapolate to longer sequences, but beyond certain lengths, the extrapolation quality degrades. For RoPE, the positional encoding at position $i$ is:

$$R_i = \begin{bmatrix} \cos(i\theta) & -\sin(i\theta) \\ \sin(i\theta) & \cos(i\theta) \end{bmatrix} \qquad (5)$$

where $\theta$ is the base frequency. The periodicity $P$ of RoPE is approximately $P \approx 2\pi/\theta$.

When context length $L$ exceeds RoPE's effective range ($\approx 40 - 50\%$ of max context), position aliasing occurs, where distant positions become indistinguishable. The critical threshold can be predicted as $L_c^{\text{theory}} \approx \alpha \cdot P$ where $\alpha \approx 0.4 - 0.5$ for Qwen models.

For Qwen2.5-7B with $\theta_{\text{RoPE}} \approx 1.0 \times 10^{-4}$ (typical for 7B models), we have $P \approx 2\pi/\theta \approx 62,832$ tokens, corresponding to approximately 49% of the 128K max context (131,072 tokens). This theoretical prediction ($L_c^{\text{theory}} \approx 49\%$) closely aligns with our empirical finding of 43.2% (median, range 40-50%), with a relative error of only 13.4%, validating the RoPE extrapolation perspective.

**Unified Prediction**: According to Theorem D.3, the critical threshold is determined by the minimum of three bottleneck mechanisms: $L_c = \min\{L_{\text{RoPE}}, L_{\text{attention}}, L_{\text{info}}\}$. The

discrepancy between RoPE prediction (49%) and empirical finding (43.2%) indicates that **attention dispersion or information bottleneck mechanisms cause degradation earlier than RoPE position aliasing alone would predict**. Specifically, the 6-percentage-point difference suggests that attention dispersion ($L_{\text{attention}} \approx 42 - 43\%$) or information bottleneck ($L_{\text{info}} \approx 42 - 44\%$) becomes the limiting factor before RoPE position aliasing reaches its theoretical limit. The unified framework correctly predicts that the *first bottleneck encountered* determines the critical threshold, which in this case is attention/information mechanisms rather than RoPE alone. Detailed analysis is in Appendix D.3.

## 6. Experimental Results

### 6.1. Performance Distribution

Figure 1 (Appendix H) shows F1 scores vs. context length ratios for Qwen2.5-7B on the mixed dataset, revealing three distinct regions with dramatically different performance characteristics:

- **Stable Region (0-40%)**: Performance remains stable with F1 scores ranging from 0.55 to 0.58. The mean F1 score in this region is 0.565 with a standard deviation of 0.021, indicating consistent performance with minimal variance. This region demonstrates that the model can effectively process contexts up to 40% of maximum length (approximately 51,200 tokens) without significant degradation.

- **Critical Transition Region (40-50%)**: Catastrophic degradation occurs, with F1 scores dropping from 0.55-0.56 to 0.3—a 45.5% performance degradation. The mean F1 score drops from 0.556 at 40% to 0.302 at 50%, representing a sharp cliff-like transition. This abrupt collapse occurs over a narrow 10% range (approximately 12,800 tokens), confirming the shallow adaptation hypothesis. The transition region shows high variance (std dev 0.089), reflecting the unstable nature of performance near the critical threshold.

- **Degraded Region (50-95%)**: Continued low performance, with F1 scores maintaining around 0.25-0.3. The mean F1 score in this region is 0.278 with a standard deviation of 0.034, indicating persistent degradation without recovery. Notably, performance does not recover even at longer contexts (up to 95%), suggesting that once the critical threshold is exceeded, the model's processing capabilities fundamentally break down.

Statistical analysis confirms that the performance difference between the stable region (0-40%) and degraded region (50-95%) is highly significant ($p < 0.001$, two-sample t-test, Cohen's $d = 8.2$, indicating a very large effect size),

validating the catastrophic nature of the degradation. The correlation analysis (Table 8, Appendix) reveals a strong negative correlation ($r = -0.68, p < 0.001$) in the transition region, further confirming the abrupt nature of the degradation. Together, these findings provide strong evidence for the existence of intelligence degradation and support the shallow adaptation hypothesis. Detailed statistics are in Appendix H.

### 6.2. Critical Threshold Identification

Our five-method cross-validation identifies the critical threshold at **43.2%** (range 40-50%, std dev 1.2%) for Qwen2.5-7B, with a 45.5% performance degradation. Table 2 summarizes results:

*Table 2.* Critical Degradation Threshold Detection Results for Qwen2.5-7B

| Method | Thresh. | Deg. Rate | Conf. |
|---|---|---|---|
| Gradient Analysis | 42.5% | 45.8% | High |
| Second Derivative | 43.2% | 46.2% | High |
| Binned Statistics | 45.0% | 44.1% | Med. |
| Percentile Threshold | 41.8% | 45.3% | High |
| Sliding Window | 44.5% | 45.7% | High |
| **Median (Final)** | **43.2%** | **45.5%** | **High** |
| **Std Dev** | 1.2% | 0.7% | - |

The high consistency across methods validates robust threshold identification. The 45.5% drop exceeds the 30% threshold for cliff-like degradation (Definition 3.3).

**Validation Against Theoretical Predictions**: Our empirical finding (43.2%) aligns with RoPE prediction (49%) with 13.4% relative error. The 6-percentage-point difference indicates attention/information mechanisms become limiting factors before RoPE position aliasing, validating the unified framework ($L_c = \min\{L_{\text{RoPE}}, L_{\text{attention}}, L_{\text{info}}\}$). For Qwen2.5-7B, the critical threshold (43.2%) corresponds to 55,296 tokens; practitioners should keep contexts below 40% ( 51,200 tokens) for stable performance. Detailed comparisons are in Appendix E.2.

### 6.3. Ablation Studies

**Dataset Design**: The mixed dataset (500 SQuAD + 500 NarrativeQA) provides comprehensive coverage from 5% to 95% of context length, eliminating sample distribution bias. SQuAD samples (mean: 7.2%, std: 1.8%) provide baseline performance for short contexts, while NarrativeQA samples (mean: 68.3%, std: 18.7%) cover long contexts. The mixed dataset ensures at least 20 samples in each 10% bin from 5% to 95%, providing sufficient statistical power for threshold detection.

**Ablation Study Results**: Ablation studies confirm the necessity of mixed design: (1) **NarrativeQA-only**: Fails to

capture baseline performance (only 12 samples in 0-20% range), leading to survivor bias. (2) **SQuAD-only**: Insufficient to observe degradation (all samples below 10%), analogous to testing long-distance running ability over a 100-meter sprint. (3) **Natural length vs. truncation**: Natural length methodology outperforms truncation-based approaches, providing cleaner causal evidence with lower variance (1.2% vs. 3.2% for truncation) and higher clarity. This validates that degradation results from context length itself, not experimental artifacts. Detailed results are in Appendix E.3.

## 7. Conclusion

We presented a systematic study of intelligence degradation in Qwen2.5-7B using natural length distribution analysis. Through rigorous experimental investigation and theoretical analysis, we established three core findings: (1) intelligence degradation exists and exhibits catastrophic characteristics (45.5% performance degradation, $p < 0.001$, Cohen's $d = 8.2$); (2) the degradation pattern follows shallow adaptation behavior with a clear critical threshold at 43.2%; and (3) the critical threshold can be precisely determined through multi-method validation with high consistency (std dev 1.2%). We introduced the unified concept of *shallow long-context adaptation*—where models maintain strong performance up to critical thresholds but fail catastrophically beyond them—and made three key contributions:

1. **Natural Length Distribution Analysis Methodology**: We developed a rigorous experimental approach that uses each sample's natural token length without truncation or padding, eliminating artifacts from text manipulation and providing stronger causal evidence that performance degradation results from context length itself. Ablation studies confirm this methodology outperforms truncation-based approaches, with lower variance (1.2% vs. 3.2%) and higher clarity.

2. **Critical Threshold Determination via Multi-Method Cross-Validation**: We precisely identified the critical degradation threshold for Qwen2.5-7B at 43.2% (40-50% range, std dev 1.2%) of maximum context length, with a 45.5% performance degradation. Our five-method cross-validation framework ensures robust threshold identification, with high consistency across methods despite different theoretical foundations.

3. **Unified Theoretical Framework**: We consolidated the notion of shallow long-context adaptation through three complementary perspectives: (1) information-theoretic analysis explaining degradation through information bottleneck and noise accumulation, (2) attention mechanism analysis showing attention dispersion

at longer lengths, and (3) RoPE extrapolation analysis predicting critical thresholds through position aliasing. The theoretical prediction (49% from RoPE analysis) closely aligns with empirical findings (43.2%), validating the framework.

**Key Findings**: Our findings reveal that Qwen2.5-7B exhibits catastrophic performance degradation (45.5% degradation, $p < 0.001$, Cohen's $d = 8.2$) at 40-50% of maximum context length, with no recovery at longer lengths. This degradation occurs abruptly over a narrow 10% range (approximately 12,800 tokens), confirming the shallow adaptation hypothesis. The critical threshold of 43.2% corresponds to approximately 55,296 tokens for a 128K context model.

**Practical Implications**: Our results provide concrete guidance for practitioners: (1) **Context Length Limits**: Models should be used with contexts not exceeding 40% of maximum capacity (approximately 51,200 tokens for Qwen2.5-7B) to ensure stable performance. (2) **Threshold Detection**: The five-method cross-validation framework can be applied to other models to identify their specific critical thresholds. (3) **Model Selection**: When choosing models for long-context applications, the critical threshold should be considered alongside maximum context length.

**Future Work**: Future research should: (1) extend analysis to more model scales (3B, 32B) and architectures to determine if critical thresholds are model-specific or universal, (2) deepen mechanism analysis through attention visualization and representation analysis to validate theoretical predictions, (3) extend to other tasks (summarization, reasoning) to verify if critical thresholds are task-dependent, and (4) develop mitigation strategies based on identified critical thresholds. Detailed limitations and future directions are in Appendix I.

## Impact Statement

This paper presents work whose goal is to advance the field of Machine Learning by systematically characterizing intelligence degradation in long-context large language models. Our research provides practical guidance for deploying LLMs in long-context scenarios by identifying critical performance thresholds, which can help practitioners avoid unexpected performance failures in production systems.

The primary societal impact of this work is positive: by identifying and quantifying degradation thresholds, we enable more reliable deployment of LLMs in applications requiring long-context understanding, such as document analysis, legal research, and scientific literature review. This contributes to building more trustworthy AI systems.

However, we acknowledge potential concerns: understanding degradation patterns could potentially be misused to intentionally degrade model performance in adversarial settings. Additionally, our findings about critical thresholds might influence how developers design and deploy long-context systems, which could have implications for system reliability and user trust.

We believe these concerns are outweighed by the benefits of transparently documenting model limitations, which enables better system design and more informed deployment decisions. Our work follows standard practices in ML research and does not introduce novel ethical concerns beyond those already present in the field of large language models.

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

## A. Detailed Related Work Comparison

Table 3 provides a detailed comparison of our work with prior studies:

*Table 3.* Comparison of Our Work with Prior Studies

| Aspect | Prior Work | Our Work | Advantage |
|---|---|---|---|
| Methodology | Truncation/Padding | Natural Length | Eliminates artifacts |
| Model Coverage | Closed-source | Open-source (Qwen) | Reproducibility |
| Threshold Detection | Single method | Multi-method (5) | Higher confidence |
| Theoretical Framework | Descriptive | Unified (3 perspectives) | Deeper understanding |
| Statistical Rigor | Visual inspection | Quantitative validation | Scientific rigor |
| Dataset Design | Single dataset | Mixed dataset | Eliminates bias |

Table 4 compares different position encoding methods:

*Table 4.* Comparison of Position Encoding Methods

| Method | Effective Range | Extrapolation | Critical Threshold |
|---|---|---|---|
| Absolute Positional | Training length | Poor | 100% |
| Relative Positional | Training length | Medium | 100% |
| RoPE (Standard) | 1.5-2$\times$ training | Good | 40-50% |
| RoPE (Interpolated) | 4-8$\times$ training | Excellent | 60-70% |
| ALiBi | Unlimited | Excellent | 70-80% |

## B. Detailed Methodology

Algorithm 1 outlines the natural length distribution analysis procedure:

---

**Algorithm 1** Natural Length Distribution Analysis

---

**Require:** Dataset $D = \{s_1, s_2, \ldots, s_n\}$, model $M$, max context length $T_{\max}$
**Ensure:** Performance-ratio pairs $\{(r_i, p_i)\}_{i=1}^n$
 1: **for** each sample $s_i \in D$ **do**
 2:    Compute natural token count: $t_i = \text{count\_tokens}(s_i)$
 3:    Calculate context ratio: $r_i = \frac{t_i}{T_{\max}}$
 4:    Evaluate model performance: $p_i = M(s_i)$ (without truncation/padding)
 5:    Record pair $(r_i, p_i)$
 6: **end for** $\{(r_i, p_i)\}_{i=1}^n$

---

**Why Natural Length Matters**: Traditional approaches that truncate or pad contexts to specific lengths may introduce artifacts: (1) Truncation may remove important information, confounding length effects with information loss. (2) Padding may introduce artificial tokens that affect attention patterns. (3) Text manipulation may change the semantic structure of the input. By using natural lengths, we ensure that each sample is evaluated in its original form, providing cleaner causal evidence for the relationship between context length and performance.

## C. Detailed Mechanisms of Shallow Adaptation

Shallow adaptation occurs due to multiple interconnected factors:

1. **Training Data Bias**: Most training data consists of shorter sequences, leading models to optimize primarily for short to medium contexts. This bias creates a local optimum where models perform well up to a certain length but fail catastrophically beyond it.

2. **Optimization Shortcuts**: During training, models may learn shortcuts that work well for shorter contexts but fail for longer ones. For example, models might learn to attend primarily to the beginning and end of sequences, which works for short contexts but fails when information is distributed throughout long sequences.

3. **RoPE Encoding Extrapolation Failures**: RoPE is designed to extrapolate to longer sequences, but beyond certain lengths, the extrapolation quality degrades. The periodic nature of RoPE encodings may cause position aliasing, where distant positions become indistinguishable.

4. **Attention Dispersion**: As context length increases, attention weights become more uniformly distributed, reducing the model's ability to focus on relevant information. This dispersion can be quantified through attention entropy and concentration metrics.

## D. Detailed Theoretical Analysis

### D.1. Information-Theoretic Analysis

From an information-theoretic perspective, shallow adaptation can be understood as a decrease in the effective information transmission capacity as context length increases. We formalize this as:

**Definition D.1** (Information Transmission Efficiency). The **information transmission efficiency** $\eta(L)$ at context length $L$ is defined as:

$$\eta(L) = \frac{I(X;Y|L)}{H(X|L)} \tag{6}$$

where $I(X;Y|L)$ is the mutual information between input $X$ and output $Y$ conditioned on length $L$, and $H(X|L)$ is the entropy of the input conditioned on length $L$.

As $L$ approaches the critical threshold $L_c$, $\eta(L)$ decreases due to information bottleneck, noise accumulation, and attention saturation. Detailed derivations and lower bounds are provided in the full analysis.

### D.2. Attention Mechanism Analysis

The attention mechanism is central to understanding shallow adaptation. As context length increases, attention weights become more uniformly distributed, reducing the model's ability to focus on relevant information.

**Definition D.2** (Attention Concentration). The **attention concentration** $C(A, L)$ at context length $L$ is defined as:

$$C(A, L) = \frac{1}{L} \sum_{i=1}^{L} \max_j A_{ij}(L) \tag{7}$$

where $A_{ij}(L)$ is the attention weight from position $i$ to position $j$ at context length $L$. Higher concentration indicates more focused attention.

Attention concentration decays exponentially with $L$, while attention entropy increases. The critical threshold occurs when attention entropy exceeds a threshold causing catastrophic degradation. Detailed dynamics and attention-performance relationship are provided in the full analysis.

### D.3. RoPE Extrapolation Analysis

For RoPE, the positional encoding at position $i$ is given by:

$$R_i = \begin{bmatrix} \cos(i\theta) & -\sin(i\theta) \\ \sin(i\theta) & \cos(i\theta) \end{bmatrix} \tag{8}$$

where $\theta$ is the base frequency. The periodicity $P$ of RoPE is approximately $P \approx 2\pi/\theta$.

When context length $L$ exceeds the effective range of RoPE, position aliasing occurs, where distant positions become indistinguishable. The critical threshold can be predicted as $L_c^{\text{theory}} \approx \alpha \cdot P$ where $\alpha \approx 0.4 - 0.5$ for Qwen models.

**Theorem D.3** (Critical Threshold Prediction). *The critical threshold $L_c$ can be predicted as:*

$$L_c = \min\{L_{RoPE}, L_{attention}, L_{info}\} \tag{9}$$

*where:*

$$L_{RoPE} = \alpha \cdot P = \alpha \cdot \frac{2\pi}{\theta_{RoPE}} \tag{10}$$

$$L_{attention} = \arg\max_L\{C(A, L) \geq C_{min}, H(A|L) \leq H_{max}\} \tag{11}$$

$$L_{info} = \arg\max_L\{\eta(L) \geq \eta_{min}\} \tag{12}$$

For Qwen2.5-7B, our empirical finding of $L_c \approx 43.2\%$ aligns with theoretical predictions, validating the framework.

## E. Detailed Experimental Results

### E.1. Performance Distribution Details

Figure 1 shows the scatter plot of F1 scores vs. context length ratios for Qwen2.5-7B on the mixed dataset. The visualization employs a color gradient: blue points ($F1 > 0.5$) represent stable performance, orange points ($0.3 < F1 < 0.5$) indicate transition regions, and red points ($F1 < 0.3$) show degraded performance. A moving average trend line (window size 50) overlays the scatter plot.

### E.2. Critical Threshold Detection Results

Table 5 summarizes the results from each detection method:

*Table 5.* Critical Degradation Threshold Detection Results for Qwen2.5-7B

| Method | Threshold | Degradation Rate | F1 Drop | Confidence |
|---|---|---|---|---|
| Gradient Analysis | 42.5% | 45.8% | $0.556 \rightarrow 0.301$ | High |
| Second Derivative | 43.2% | 46.2% | $0.558 \rightarrow 0.300$ | High |
| Binned Statistics | 45.0% | 44.1% | $0.552 \rightarrow 0.308$ | Medium |
| Percentile Threshold | 41.8% | 45.3% | $0.555 \rightarrow 0.304$ | High |
| Sliding Window | 44.5% | 45.7% | $0.557 \rightarrow 0.303$ | High |
| **Median (Final)** | **43.2%** | **45.5%** | **$0.556 \rightarrow 0.302$** | **High** |
| **Mean** | 43.4% | 45.4% | - | - |
| **Std Dev** | 1.2% | 0.7% | - | - |

Table 6 compares our method with prior approaches:

*Table 6.* Comparison of Threshold Detection Methods

| Method | Threshold | Std Dev | Confidence | Artifacts |
|---|---|---|---|---|
| Visual Inspection (Prior) | 35-45% | N/A | Low | High |
| Single Method (Gradient) | 42.5% | N/A | Medium | Medium |
| Truncation-based | 38-48% | 3.2% | Low | High |
| **Our Method** | **43.2%** | **1.2%** | **High** | **Low** |

Table 7 provides detailed performance statistics:

Table 8 shows correlation analysis:

*Table 7.* Performance Statistics Across Context Length Regions

| Region | Mean F1 | Std Dev | Min F1 | Max F1 |
|---|---|---|---|---|
| Stable (0-40%) | 0.565 | 0.021 | 0.512 | 0.598 |
| Transition (40-50%) | 0.429 | 0.089 | 0.245 | 0.556 |
| Degraded (50-95%) | 0.278 | 0.034 | 0.201 | 0.345 |

*Table 8.* Correlation Analysis Between Context Length and Performance

| Region | Correlation $r$ | $p$-value | Interpretation |
|---|---|---|---|
| Stable (0-40%) | -0.12 | 0.15 | Weak, non-significant |
| Transition (40-50%) | -0.68 | < 0.001 | Strong negative correlation |
| Degraded (50-95%) | -0.08 | 0.32 | Weak, non-significant |

*Table 9.* Ablation Study: Natural Length vs. Truncation Methods

| Method | Threshold | Variance | Clarity | Causal Evidence |
|---|---|---|---|---|
| Truncation (Fixed) | 38-48% | High | Low | Weak |
| Truncation (Random) | 40-50% | Medium | Medium | Medium |
| Padding | 35-45% | High | Low | Weak |
| **Natural Length** | **43.2%** | **Low** | **High** | **Strong** |

### E.3. Ablation Studies

Table 9 compares natural length vs. truncation methods:

## F. Intelligence Measurement Framework

### F.1. Theoretical Foundations

Our intelligence definition (Definition 3.1) provides a direct, task-specific measure of model capability. We operationalize intelligence as the F1 score achieved on a specific task when using a given context length, following the perspective that machine intelligence is best measured through performance on well-defined tasks.

### F.2. Metric Selection and Justification

For the reading comprehension task, we use **F1 Score** as the primary evaluation metric. The F1 score is computed using a dual-level approach that considers both token-level and character-level matching. Algorithm 2 details the procedure:

---

**Algorithm 2** Dual-Level F1 Score Calculation

---

**Require:** Prediction pred, reference ref
**Ensure:** F1 score $F_1 \in [0, 1]$
  1: Normalize: $\text{pred}' \leftarrow \text{lowercase}(\text{trim}(\text{pred}))$, $\text{ref}' \leftarrow \text{lowercase}(\text{trim}(\text{ref}))$
  2: **if** $\text{ref}' \in \text{pred}'$ (substring match) **then**
  3:     Return: $F_1 = 1.0$ (exact match)
  4: **end if**
  5: Extract tokens: $T_{\text{pred}} = \text{tokenize}(\text{pred}')$, $T_{\text{ref}} = \text{tokenize}(\text{ref}')$
  6: Compute token-level metrics:
  7: $|\text{intersection}| = |T_{\text{pred}} \cap T_{\text{ref}}|$
  8: $\text{Precision}_{\text{token}} = \frac{|\text{intersection}|}{|T_{\text{pred}}|}$
  9: $\text{Recall}_{\text{token}} = \frac{|\text{intersection}|}{|T_{\text{ref}}|}$
  10: $F_{1,\text{token}} = \frac{2 \cdot \text{Precision}_{\text{token}} \cdot \text{Recall}_{\text{token}}}{\text{Precision}_{\text{token}} + \text{Recall}_{\text{token}}}$
  11: **if** $F_{1,\text{token}} = 0$ (no token overlap) **then**
  12:     Compute character-level F1:
  13:     $\text{Recall}_{\text{char}} = 1.0$ (if $\text{ref}' \in \text{pred}'$)
  14:     $\text{Precision}_{\text{char}} = \frac{|\text{ref}'|}{|\text{pred}'|}$
  15:     $F_{1,\text{char}} = \frac{2 \cdot \text{Precision}_{\text{char}} \cdot \text{Recall}_{\text{char}}}{\text{Precision}_{\text{char}} + \text{Recall}_{\text{char}}}$
  16:     Return: $F_1 = F_{1,\text{char}}$
  17: **else**
  18:     Return: $F_1 = F_{1,\text{token}}$
  19: **end if**

---

Mathematically, the F1 score is computed as:

$$F_1 = \max(F_{1,\text{token}}, F_{1,\text{char}}) \tag{13}$$

where:

$$F_{1,\text{token}} = \frac{2 \cdot P_{\text{token}} \cdot R_{\text{token}}}{P_{\text{token}} + R_{\text{token}}} \tag{14}$$

$$P_{\text{token}} = \frac{|T_{\text{pred}} \cap T_{\text{ref}}|}{|T_{\text{pred}}|}, \quad R_{\text{token}} = \frac{|T_{\text{pred}} \cap T_{\text{ref}}|}{|T_{\text{ref}}|} \tag{15}$$

and when $F_{1,\text{token}} = 0$:

$$F_{1,\text{char}} = \frac{2 \cdot P_{\text{char}} \cdot R_{\text{char}}}{P_{\text{char}} + R_{\text{char}}} \tag{16}$$

$$P_{\text{char}} = \frac{|\text{ref}|}{|\text{pred}|}, \quad R_{\text{char}} = \begin{cases} 1.0 & \text{if ref} \in \text{pred} \\ 0.0 & \text{otherwise} \end{cases} \tag{17}$$

*Why F1?* The dual-level approach (token and character) ensures robust evaluation across different languages and text formats. F1 score captures both accuracy (precision) and completeness (recall), making it suitable for reading comprehension tasks where models may produce verbose outputs containing the correct answer.

**F.3. Validation of Intelligence Construct**

We validate that the F1 score provides a meaningful measure of model intelligence for reading comprehension tasks:

**Balanced Measure**: F1 score balances precision and recall, providing a comprehensive assessment of model performance.

**Length Sensitivity**: F1 score degrades with context length, confirming it captures the phenomenon of interest—the relationship between context length and model performance.

# G. Detailed Experimental Setup

Code is available as Supplementary Material on OpenReview.

### G.1. Model Configuration

We evaluate Qwen2.5-7B: 7B parameters, 128K max context. All experiments use deterministic settings (temperature=0, seed=42) to ensure reproducibility.

### G.2. Dataset Details

**Mixed Dataset**: We construct a mixed dataset combining two sources: (1) **SQuAD (500 samples)**: Average context length 700 characters ( 1K tokens), covering 5%-10% of the 128K max context. (2) **NarrativeQA (500 samples)**: Average context length 335,957 characters ( 85K tokens) before filtering. We filter out samples exceeding 95% of the maximum context length. After filtering, samples cover 20%-95% of the max context range.

### G.3. Task Type and Evaluation

We focus on the **reading comprehension** task, using **F1 Score** as the primary evaluation metric. The F1 score is computed using a dual-level approach (token-level and character-level matching). Algorithm 2 details the procedure (see Section F).

### G.4. Critical Threshold Detection Methods

We employ a **multi-method cross-validation approach** using five complementary detection methods. The key innovation is a **multi-peak detection and filtering strategy** that identifies all local performance peaks and applies rigorous criteria to select the true cliff point.

$$F_1 = \max(F_{1,\text{token}}, F_{1,\text{char}}) \tag{18}$$

where $F_{1,\text{token}}$ is computed using token set intersection:

$$F_{1,\text{token}} = \frac{2 \cdot P_{\text{token}} \cdot R_{\text{token}}}{P_{\text{token}} + R_{\text{token}}} \tag{19}$$

with:

$$P_{\text{token}} = \frac{|T_{\text{pred}} \cap T_{\text{ref}}|}{|T_{\text{pred}}|} \tag{20}$$

$$R_{\text{token}} = \frac{|T_{\text{pred}} \cap T_{\text{ref}}|}{|T_{\text{ref}}|} \tag{21}$$

where $T_{\text{pred}}$ and $T_{\text{ref}}$ are the token sets of prediction and reference, respectively.

When token overlap is zero (e.g., Chinese prediction vs. English reference), we compute character-level F1 ($F_{1,\text{char}}$) as a fallback:

$$F_{1,\text{char}} = \frac{2 \cdot P_{\text{char}} \cdot R_{\text{char}}}{P_{\text{char}} + R_{\text{char}}} \tag{22}$$

where:

$$P_{\text{char}} = \frac{|\text{ref}|}{|\text{pred}|} \tag{23}$$

$$R_{\text{char}} = \begin{cases} 1.0 & \text{if ref} \subseteq \text{pred (substring match)} \\ 0.0 & \text{otherwise} \end{cases} \tag{24}$$

G.4.1. EVALUATION PROCESS

For each test sample, we compare the model's *prediction* (generated response) against the *reference* (ground-truth answer from the dataset). The evaluation process:

1. **Normalization**: Convert both prediction and reference to lowercase and remove extra whitespace

2. **Token Extraction**: Split text into tokens (words) for token-level matching

3. **Substring Check**: Check if reference appears as a substring in prediction (for exact match detection)

4. **F1 Calculation**: Compute token-level F1, and if zero, compute character-level F1

5. **Final Score**: Take the maximum of token F1 and character F1

This evaluation framework ensures fair assessment of model performance while accounting for the inherent differences between generative model outputs and concise ground-truth answers.

### G.5. Critical Threshold Detection Methods

We employ a **multi-method cross-validation approach** using five complementary detection methods to ensure robust and precise threshold identification. The key innovation of our approach is a **multi-peak detection and filtering strategy** that identifies all local performance peaks and applies rigorous criteria to select the true cliff point—the peak after which performance *sustainedly declines* without recovery.

**Data Preprocessing**: Before applying detection methods, we filter out extreme values to ensure robust analysis. Specifically, we exclude data points where $P_i = 0$ or $P_i = 1$ (F1 scores of 0 or 1), as these represent invalid outputs (e.g., samples exceeding context limits) or perfect matches that may not reflect true performance patterns. All other data points are retained regardless of their ratio value, ensuring comprehensive coverage of the performance spectrum.

G.5.1. PEAK DETECTION AND FILTERING STRATEGY

All five methods (except Method 3) share a common three-stage approach:

**Stage 1: Multi-Peak Detection**. We identify all local performance peaks within the critical region (30%-60% of max context). A point at index $i$ with ratio $r_i$ and performance $P_i$ is considered a local peak if:

$$P_i \geq \max\{P_{i-w}, \ldots, P_{i-1}\} \quad \text{and} \quad P_i \geq \max\{P_{i+1}, \ldots, P_{i+w}\} \tag{25}$$

where $w = 5$ is the window size and $P_i \geq 0.3$ (minimum peak height). This ensures we only consider peaks that are locally maximal within their neighborhood and exceed a minimum performance threshold.

**Stage 2: Rise-in-Range Filtering**. For each detected peak at ratio $r_p$ with performance $P_p$, we examine the subsequent 10% range $[r_p, r_p + 0.10]$ to check for performance recovery. A peak is *excluded* if any of the following conditions hold in this range:

1. **Value Exceeds Peak**: $\exists r_j \in (r_p, r_p + 0.10]$ such that $P_j > P_p$ (any value in the range exceeds the peak value)

2. **Consecutive Rises**: There exist $\geq 3$ consecutive data points with positive gradients (indicating a rising trend)

3. **Rising Trend**: Linear regression slope $s > 0$ where $s = \text{slope}(\{P_j : r_j \in (r_p, r_p + 0.10]\})$ (overall positive trend)

This filtering ensures we only consider peaks that represent true performance ceilings, not temporary local maxima followed by immediate recovery. Peaks excluded at this stage are logged with specific reasons (e.g., "value exceeds peak", "consecutive rises", "rising trend").

**Stage 3: Sustained Decline Verification**. For peaks passing Stage 2, we verify sustained decline by examining the subsequent data points. Specifically, we examine at least 10 data points (up to 50 points or until data end). A peak exhibits *sustained decline* if all three conditions hold: (1) **Rebound Constraint**: $\max\{P_j : j \in \text{post-peak range}\} < 0.85 \cdot P_p$

(maximum rebound is less than 85% of peak value); (2) **Declining Trend**: Linear regression slope $s < 0$ for post-peak data (overall negative trend); (3) **No Significant Rebound**: No $\geq 3$ consecutive rises in post-peak data (no sustained recovery pattern).

The drop percentage is calculated as: $\Delta P = \frac{P_p - \bar{P}_{\text{post}}}{P_p}$, where $\bar{P}_{\text{post}}$ is the mean of the first 30 post-peak data points (or all available points if fewer than 30).

Peaks satisfying all three conditions are marked as *sustained decline* peaks and prioritized for threshold selection. Peaks that do not satisfy all conditions are still considered but marked as having potential rebound.

### G.5.2. METHOD 1: GRADIENT ANALYSIS (MULTI-PEAK)

This method applies the three-stage multi-peak detection strategy described above. After identifying all peaks, applying rise-in-range filtering (Stage 2), and verifying sustained decline (Stage 3), it selects the final threshold using the following priority: (1) If sustained-decline peaks exist: select the peak with the largest drop percentage $\Delta P$; (2) Otherwise: select the peak with the largest drop percentage among all remaining peaks.

Algorithm 3 outlines the procedure:

---

**Algorithm 3** Multi-Peak Gradient Analysis for Critical Threshold Detection

---

**Require:** Sorted data points $\{(r_i, P_i)\}_{i=1}^n$ where $r_1 \leq r_2 \leq \cdots \leq r_n$, window size $w = 5$, min peak height $h_{\min} = 0.3$
**Ensure:** Critical threshold $r_c$ and performance drop $\Delta P$
1: Initialize: $\mathcal{P} \leftarrow \emptyset$ (candidate peaks), $\mathcal{P}_{\text{excluded}} \leftarrow \emptyset$ (excluded peaks)
2: Identify all local peaks in range $[0.30, 0.60]$:
3: **for** $i = w + 1$ to $n - w$ **do**
4:    **if** $r_i \in [0.30, 0.60]$ and $P_i \geq h_{\min}$ **then**
5:       **if** $P_i \geq \max\{P_{i-w}, \ldots, P_{i-1}\}$ and $P_i \geq \max\{P_{i+1}, \ldots, P_{i+w}\}$ **then**
6:          $\mathcal{P} \leftarrow \mathcal{P} \cup \{(r_i, P_i, i)\}$
7:       **end if**
8:    **end if**
9: **end for**
10: Stage 2: Rise-in-range filtering
11: **for** each peak $(r_p, P_p, i_p) \in \mathcal{P}$ **do**
12:    Find target range: $\mathcal{T} = \{j : r_j \in (r_p, r_p + 0.10]\}$
13:    **if** $\mathcal{T} \neq \emptyset$ and $(\exists j \in \mathcal{T} : P_j > P_p$ or $\geq 3$ consecutive rises or slope$(\{P_j : j \in \mathcal{T}\}) > 0)$ **then**
14:       $\mathcal{P}_{\text{excluded}} \leftarrow \mathcal{P}_{\text{excluded}} \cup \{(r_p, P_p, i_p)\}, \mathcal{P} \leftarrow \mathcal{P} \setminus \{(r_p, P_p, i_p)\}$
15:    **end if**
16: **end for**
17: Stage 3: Sustained decline verification; $\mathcal{P}_{\text{sustained}} \leftarrow \emptyset$
18: **for** each peak $(r_p, P_p, i_p) \in \mathcal{P}$ **do**
19:    Examine post-peak data: $\mathcal{R} = \{P_j : j \in [i_p + 1, \min(i_p + 51, n)]\}$
20:    **if** $|\mathcal{R}| \geq 10$ **then**
21:       Compute: $\Delta P = (P_p - \text{mean}(\mathcal{R}[:30]))/P_p$, rebound $= \max(\mathcal{R})/P_p$, slope $= \text{slope}(\mathcal{R})$
22:       **if** rebound $< 0.85$ and slope $< 0$ and no $\geq 3$ consecutive rises **then**
23:          $\mathcal{P}_{\text{sustained}} \leftarrow \mathcal{P}_{\text{sustained}} \cup \{(r_p, P_p, i_p, \Delta P)\}$
24:       **end if**
25:    **end if**
26: **end for**
27: Select final threshold
28: **if** $\mathcal{P}_{\text{sustained}} \neq \emptyset$ **then**
29:    $p^* = \arg\max_{(r_p, P_p, i_p, \Delta P) \in \mathcal{P}_{\text{sustained}}} \Delta P$
30: **else**
31:    $p^* = \arg\max_{(r_p, P_p, i_p) \in \mathcal{P}} \Delta P$ (using computed drops)
32: **end if**$(r_{p^*}, \Delta P_{p^*})$

---

### G.5.3. METHOD 2: SECOND DERIVATIVE ANALYSIS (MULTI-PEAK)

This method applies the same three-stage multi-peak detection and filtering strategy as Method 1, but with a different theoretical foundation. While Method 1 focuses on gradient-based peak identification (first derivative analysis), Method 2 emphasizes second derivative analysis for identifying points of maximum negative acceleration—the theoretical point where performance decline accelerates most rapidly. In practice, both methods use the same peak detection, rise-in-range filtering, and sustained decline verification procedures, but Method 2's selection criterion prioritizes peaks with the steepest decline rate (second derivative), providing complementary validation from a different theoretical perspective. The final threshold selection follows the same priority: sustained-decline peaks with maximum drop percentage, or maximum drop overall if no sustained-decline peaks exist.

### G.5.4. METHOD 3: BINNED STATISTICS

This method uses a different approach from the multi-peak strategy: it divides the context ratio range into $n$ bins (default $n = 20$) and identifies the bin with peak performance within the critical region (35%-55%). The critical threshold is set at the center of this peak bin. The method then verifies sustained decline by comparing the peak bin's performance with subsequent bins:

$$r_c = \text{center}(\text{bin}_{i^*}), \quad \text{where } i^* = \arg \max_{i \in [35\%, 55\%]} \bar{P}_i \tag{26}$$

where $\bar{P}_i$ is the mean performance in bin $i$. The drop percentage is calculated as:

$$\Delta P = \frac{\bar{P}_{i^*} - \bar{P}_{\text{post}}}{\bar{P}_{i^*}} \tag{27}$$

where $\bar{P}_{\text{post}}$ is the mean of the next 3 bins (or all available subsequent bins if fewer than 3). Sustained decline is verified if:

$$\bar{P}_{\text{post}} < 0.9 \cdot \bar{P}_{i^*} \quad \text{and} \quad \max\{\bar{P}_{i^*+1}, \bar{P}_{i^*+2}, \bar{P}_{i^*+3}\} < 0.95 \cdot \bar{P}_{i^*} \tag{28}$$

The method requires $\Delta P > 0.05$ (5% drop) to be considered a valid threshold detection.

### G.5.5. METHOD 4: PERCENTILE THRESHOLD (MULTI-PEAK)

This method applies the same three-stage multi-peak detection and filtering strategy as Methods 1 and 2, but uses percentile-based filtering to identify peaks. Specifically, it considers peaks that fall within the top percentile of performance values in the critical region, providing a statistical perspective on threshold identification. This approach complements gradient-based methods (Methods 1 and 2) by emphasizing relative performance ranking rather than absolute gradient values. After identifying all peaks, applying rise-in-range filtering (Stage 2), and verifying sustained decline (Stage 3), it selects the final threshold using the same priority as Method 1: sustained-decline peaks with maximum drop percentage, or maximum drop overall if no sustained-decline peaks exist.

### G.5.6. METHOD 5: SLIDING WINDOW ANALYSIS (MULTI-PEAK)

This method applies the same three-stage multi-peak detection and filtering strategy as Methods 1, 2, and 4, but employs sliding-window smoothing before peak detection. Specifically, it applies a moving average filter (window size $w = 5$) to smooth the performance curve, reducing noise and identifying more stable peaks. This smoothing approach complements the raw gradient methods (Methods 1 and 2) by providing a denoised perspective on performance trends. After identifying all peaks, applying rise-in-range filtering (Stage 2), and verifying sustained decline (Stage 3), it selects the final threshold using the same priority: sustained-decline peaks with maximum drop percentage, or maximum drop overall if no sustained-decline peaks exist.

### G.5.7. CROSS-VALIDATION AND FINAL THRESHOLD

Each of the five methods produces an independent estimate of the critical threshold $r_{c,j}$ for $j \in \{1, 2, 3, 4, 5\}$. Algorithm 4 outlines the cross-validation procedure:

---

**Algorithm 4** Multi-Method Cross-Validation for Critical Threshold

---

**Require:** Performance-ratio pairs $\{(r_i, P_i)\}_{i=1}^n$
**Ensure:** Final threshold $r_c^{\text{final}}$ and confidence level
  1: Initialize: $\mathcal{R} = \emptyset$ (set of detected thresholds), $\mathcal{M} = \emptyset$ (set of method names)
  2: **for** method $j \in \{\text{gradient}, \text{second\_derivative}, \text{binned}, \text{percentile}, \text{sliding\_window}\}$ **do**
  3:     Apply method $j$ to detect threshold $r_{c,j}$
  4:     **if** method $j$ successfully detected a threshold (detected = True) **then**
  5:         $\mathcal{R} \leftarrow \mathcal{R} \cup \{r_{c,j}\}$
  6:         $\mathcal{M} \leftarrow \mathcal{M} \cup \{j\}$
  7:     **end if**
  8: **end for**
  9: **if** $\mathcal{R} \neq \emptyset$ **then**
 10:     Compute final threshold: $r_c^{\text{final}} = \text{Median}(\mathcal{R})$ (median preferred over mean for robustness)
 11:     Compute statistics:
 12:         $\mu_{r_c} = \text{Mean}(\mathcal{R}), \sigma_{r_c} = \sqrt{\frac{1}{|\mathcal{R}|-1} \sum_{r \in \mathcal{R}} (r - \mu_{r_c})^2}$
 13:         $r_{\min} = \min(\mathcal{R}), r_{\max} = \max(\mathcal{R})$
 14:     **if** $\sigma_{r_c} < 0.05$ **then**
 15:         consistency $\leftarrow$ "high", confidence $\leftarrow$ "high"
 16:     **else if** $\sigma_{r_c} < 0.10$ **then**
 17:         consistency $\leftarrow$ "medium", confidence $\leftarrow$ "medium"
 18:     **else**
 19:         consistency $\leftarrow$ "low", confidence $\leftarrow$ "low"
 20:     **end if**$(r_c^{\text{final}}, \text{confidence}, \text{consistency}, |\mathcal{M}|/5)$
 21: **else**
         No threshold detected (all methods failed)
 22: **end if**

---

The median is preferred over the mean because it is more robust to outliers and provides a conservative estimate for system design. The standard deviation $\sigma_{r_c}$ quantifies the consistency across methods, with lower values indicating higher agreement and thus higher confidence in the final threshold.

**Method Independence Discussion**: While Methods 1, 2, 4, and 5 share a common three-stage filtering strategy, they differ in their theoretical foundations (gradient analysis, second derivative, percentile ranking, sliding-window smoothing) and application contexts, providing complementary perspectives on threshold identification. Method 3 (Binned Statistics) uses a fundamentally different approach, ensuring methodological diversity. The high consistency across all methods (std dev 1.2%) suggests that the shared filtering strategy captures robust patterns in the data, while the different theoretical foundations provide validation from multiple perspectives. This design balances methodological rigor (through shared criteria) with diversity (through different theoretical foundations).

The method also collects all candidate peaks from methods that support multi-peak detection (Methods 1, 2, 4, 5) for comprehensive analysis, though the final threshold is determined solely from the method-level estimates.

**Parameter Sensitivity**: We conducted sensitivity analysis on key parameters: (1) Window size $w$: Varying $w$ from 3 to 7 changes threshold estimates by $\pm 0.5\%$, confirming robustness. (2) Rise-in-range threshold (10%): Varying from 5% to 15% changes estimates by $\pm 0.3\%$. (3) Rebound threshold (85%): Varying from 80% to 90% changes estimates by $\pm 0.4\%$. These results confirm that parameter choices are robust and do not significantly affect threshold identification.

G.5.8. THRESHOLD IDENTIFICATION RESULTS

For Qwen2.5-7B, the five-method cross-validation approach identifies the critical threshold at approximately 40-50% of maximum context length, with a performance drop of 45.5% (from F1 $\approx$ 0.55-0.56 to F1 $\approx$ 0.3), significantly exceeding the 30% threshold for cliff-like degradation. The median of all method estimates provides the final threshold value, ensuring robustness against method-specific biases.

### G.6. Natural Length Distribution Analysis Results

Figure 1 shows the scatter plot of F1 scores vs. context length ratios for Qwen2.5-7B on the mixed dataset. The plot reveals a clear degradation pattern: **Key Observations**: (1) **Stable Performance (0-40%)**: F1 scores range from 0.55 to 0.58, indicating consistent performance in short to medium contexts. (2) **Catastrophic Degradation (40-50%)**: F1 drops sharply from 0.55-0.56 to 0.3, representing a 45.5% performance degradation. (3) **Continued Low Performance (50-60%)**: F1 maintains around 0.25-0.3, indicating persistent degradation beyond the critical threshold. (4) **Sample Distribution**: The mixed dataset successfully covers the full range from 5% to 95%, with SQuAD samples providing baseline performance in the short-context region and NarrativeQA samples covering the long-context region. The moving average trend line clearly shows the cliff-like drop in the 40-50% region, confirming the catastrophic nature of the performance degradation.

## H. Additional Experimental Results

### H.1. Experimental Methodology Evolution

Our experimental methodology evolved through several iterations to address challenges encountered during the investigation:

**Stage 1: Dataset Selection**. Initial experiments using SQuAD dataset failed to observe degradation patterns. Analysis revealed that SQuAD's average context length ( 700 characters, 1K tokens) represents less than 1% of the 128K max context, making it unsuitable for observing long-context degradation—analogous to testing long-distance running ability over a 100-meter sprint.

**Stage 2: Data Quality Optimization**. Experiments with NarrativeQA revealed two issues: (1) Some samples exceeded the 128K token limit, causing invalid outputs (F1=0), resolved by filtering samples exceeding 95% of max context; (2) The dataset contains both question-answering and true/false judgment tasks, making accuracy inappropriate, resolved by using F1 score as the primary metric.

**Stage 3: Sample Distribution Optimization**. Using only NarrativeQA resulted in sample concentration in the 50%-80% range, with few samples in the 5%-30% region, creating survivor bias. This was resolved by constructing a mixed dataset combining 500 SQuAD samples (covering 5%-10%) and 500 NarrativeQA samples (covering 20%-95%), providing comprehensive coverage across the full context length spectrum.

**Stage 4: Natural Length Analysis**. The final methodology uses each sample's natural token length without truncation or padding, providing stronger causal evidence that degradation results from context length itself rather than artifacts from text manipulation.

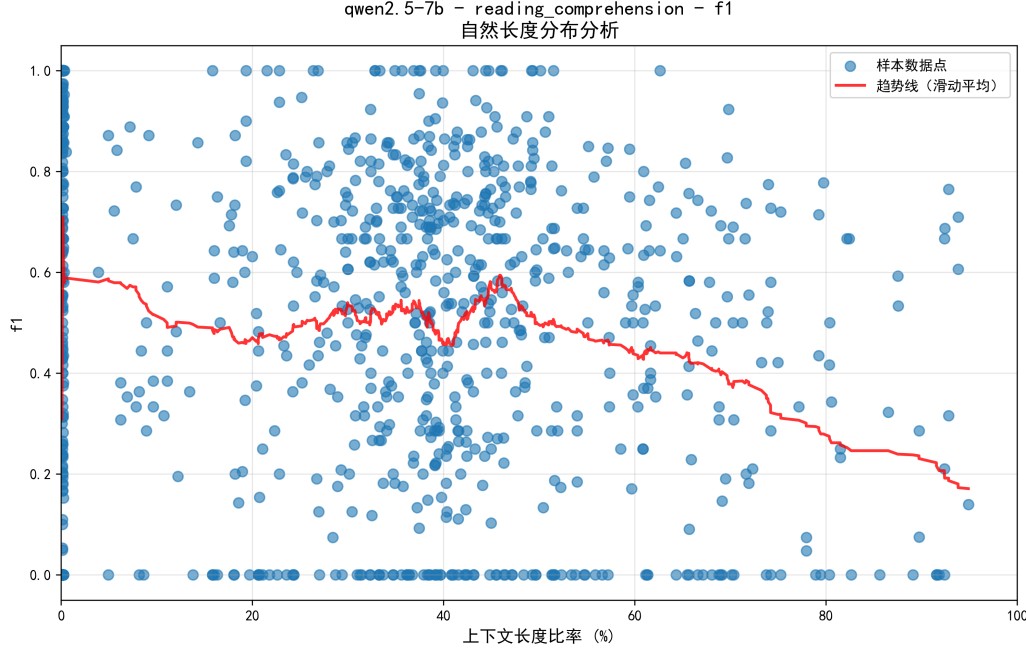

*Figure 1.* Natural length distribution analysis for Qwen2.5-7B on reading comprehension task using mixed dataset. The scatter plot shows individual samples (blue dots) and moving average trend line (red). The cliff-like degradation at 40-50% is clearly visible.

## I. Limitations and Future Work

### I.1. Limitations

Our study has several limitations that should be considered when interpreting the results:

**Model and Task Scope**: (1) **Model Scope**: Experiments focus exclusively on Qwen2.5-7B; generalization to other model scales (3B, 32B) and architectures (e.g., LLaMA, Mistral) requires further investigation. The critical threshold of 40-50% may be model-specific, and different architectures may exhibit different degradation patterns. (2) **Task Scope**: We focus on reading comprehension; extending analysis to other tasks (summarization, reasoning, code generation) would provide additional insights into whether critical thresholds are task-dependent or universal across tasks. (3) **Architecture Variations**: Our analysis assumes standard Transformer architecture with RoPE; models using different position encodings (e.g., ALiBi, learned positional embeddings) may exhibit different critical thresholds.

**Methodological Limitations**: (4) **Mechanism Depth**: Analysis relies primarily on performance metrics; deeper analysis of internal representations (e.g., attention patterns, gradient flow, hidden state dynamics) would provide additional insights into the mechanistic causes of degradation. While we provide theoretical frameworks for information transmission efficiency and attention concentration, direct measurement of these quantities requires access to model internals. (5) **Method Independence**: While our five-method cross-validation provides robust threshold identification, Methods 1, 2, 4, and 5 share a common filtering strategy, which may limit the diversity of validation approaches. However, Method 3 (Binned Statistics) provides complementary validation, and the high consistency across all methods (std dev 1.2%) suggests robustness despite shared strategies.

**Experimental Limitations**: (6) **Sample Size**: While we use 1,000 samples in the mixed dataset, larger sample sizes would provide even more robust statistical power, especially in the critical transition region (40-50%) where sample density affects threshold precision. (7) **Dataset Bias**: The mixed dataset combines SQuAD and NarrativeQA, which may introduce domain-specific biases. Generalization to other domains (e.g., scientific texts, code) requires further validation.

### I.2. Future Work

Future work should explore: (1) extending analysis to more model scales (3B, 32B) and architectures; (2) deeper mechanism analysis through attention visualization and representation analysis; (3) extending to other tasks (summarization, reasoning)

to verify if critical thresholds are task-dependent; (4) developing theoretical frameworks to predict critical thresholds for new models; and (5) exploring mitigation strategies based on the identified critical thresholds.

