# OpenReview forum: "Intelligence Degradation in Long-Context LLMs:Critical Threshold Determination via Natural Length Distribution Analysis"
_ICML.cc/2026/Conference — Submitted to ICML 2026_

### Official Review · Reviewer_qYxk · 2026-02-25

**Soundness:** 2
**Presentation:** 2
**Significance:** 2
**Originality:** 3
**Overall Recommendation:** 1
**Confidence:** 3

**Summary:**

This paper investigates intelligence degradation in long-context LLMs, focusing on a phenomenon called "shallow long-context adaptation."  The paper finds that Qwen2.5-7B maintains stable performance up to ~40% of their maximum context window, then collapses catastrophically rather than degrading gradually.

The paper utilizes a natural length distribution analysis approach (avoiding artificial truncation/padding artifacts) to identify the degradation. There are also theoretical analyses explaining why the degradation occurs, focusing on attention dispersion, RoPE embeddings, and information theory.

The experiments mainly focus on question answering datasets like SQuAD.

**Compliance With Llm Reviewing Policy:**

Affirmed.

**Final Justification:**

The rebuttal did not address my main concerns about the experimental depth of the paper. Many experiments are needed to reinforce this paper. I recommend a reject.

**Key Questions For Authors:**

How would this methodology fare for other models, like LLaMA-3.1, Gemma-3, or Olmo-3? How do results transfer on other tasks like multi-document summarization?

You utilize F1 score, but how can this analysis by utilized in open-ended generation tasks? Modern-day LLM usage is mainly on open-ended generation.

Given your definition of context ratio, what would happen if you diversify the distribution of context ratios and train the model with SFT on these datapoints? Does the degradation cliff get mitigated? What are other ways to potentially change the degradation cliff into a gradual decline in performance or maintain stability?

**Limitations:**

No. They do not mention the limited models, datasets, or empirical experiments to back up their theoretical analyses.

**Strengths And Weaknesses:**

Strengths:
The paper tackles an ongoing problem with long-context language models, namely the performance degradation as the context increases. The paper introduces a natural length distribution analysis to identify the degradation with respect to the natural token sizes of the inputs that the model is tested on.

Weaknesses:
Limited Models: The paper only focuses on Qwen-2.5-7B. It is difficult to extrapolate this analyses out to other models.

Limited Tasks: The paper only focuses on 1000 datapoints from SQuAD and NarrativeQA. It is difficult to interpret these results when long context LMs may be used for other tasks, like needle in a haystack or information extraction, information summarization, etc. The paper's experimental setup is fundamentally limited.

Limited Analyses: For the theoretical analyses, there is no empirical evidence conducted to explain why this is the case. The shallow theoretical investigations are not proven, so the analysis method is not sound.

---

> ### Author Rebuttal · Authors · 2026-03-26
>
> Thank you for the constructive feedback. We agree that the current study is limited in scope and should not be over-interpreted as a universal claim across models and tasks. We will revise the paper to better clarify what is empirically supported in this specific setting and what remains hypothesis-level.
>
> Limited model/task scope
> Our experiments focus on Qwen2.5-7B on reading comprehension with a mixed SQuAD + NarrativeQA dataset under a natural-length protocol. This allows us to characterize a sharp performance breakpoint in a controlled evaluation setting, but it does not, by itself, establish universality across architectures or downstream tasks. We will revise the abstract/Conclusion to explicitly state that our results are model- and task-conditioned, and we will emphasize that extension to models such as LLaMA/Gemma/OLMo and to tasks like multi-document summarization is a clear direction for future work (already outlined in the manuscript limitations/future work).
>
> Theoretical explanations are not empirically verified.
> We acknowledge that our information-theoretic / attention / RoPE discussions are intended as hypothesis-level motivation rather than directly proven causal mechanisms. We will revise the wording to avoid implying that these mechanisms are empirically demonstrated, and we will clarify the evidence we currently have (performance breakpoint localization) versus what requires additional measurements (e.g., internal activation/attention statistics across lengths).
>
> Use of F1 and relevance to open-ended generation.
> We use F1 because the main evaluation task is span-based question answering. The key methodological contribution is not the specific metric, but the general procedure: measure task performance as a function of effective context ratio and apply breakpoint detection to identify an “effective long-context” operating region. For open-ended generation tasks, the same protocol can be applied by replacing F1 with task-appropriate metrics (e.g., factuality/correctness, ROUGE/BERTScore, or other correctness-based evaluations). We will add a brief section clarifying how to adapt the framework to generation settings.
>
> SFT / context-ratio diversification and whether the cliff can be mitigated.
> We agree this is important. However, this paper currently reports measurement and threshold localization rather than optimization/mitigation experiments. We will clarify that changing the training distribution over context ratios (e.g., SFT on diversified length ratios) and other training/inference strategies are potential ways to shift or soften the breakpoint, and we position these as future work, rather than claiming they are resolved here.

---

> > ### Author Rebuttal · Reviewer_qYxk · 2026-04-03
> >
> > The extensions I addressed are needed to strengthen the paper significantly.

---

### Official Review · Reviewer_dRr6 · 2026-03-06

**Soundness:** 1
**Presentation:** 1
**Significance:** 2
**Originality:** 1
**Overall Recommendation:** 1
**Confidence:** 4

**Summary:**

This paper focuses on the problem of performance degradation of LLMs in long-context scenarios. Motivated by the example that LLMs experience significant performance degradation after certain token lengths, which is called intelligence degradation. Afterwards, the authors aim to determine the precise thresholds when intelligence degradation happens. To implement this, they adopt multiple approaches to analyse the relationship between performance and natural length distribution. Besides the empirical determination, they also propose theoretical framework to predict.

**Compliance With Llm Reviewing Policy:**

Affirmed.

**Final Justification:**

Since the authors haven't resolved my concerns regarding this version of paper, I will maintain my original score as strong reject. Hope the authors will follow the outlined plan to improve the quality of paper and submit to another venue.

**Key Questions For Authors:**

Please see the weaknesses.

**Limitations:**

Yes

**Strengths And Weaknesses:**

Strengths: Long context is an important scenario for LLMs. This paper may advance the research on the "effective long context".

Weaknesses: To be fair, the paper is poorly written, for example:

1. The main paper is only about 6 pages while the authors put a lot of content to the appendix, including many important results and information that may help the readers understand this paper.

2. There should be a large scale of literature about this long-context topic while the authors only cite less than 10 literature.

3. There are almost no visual illustrations to convey the idea and empirical results. The most important figure (Figure 1) about experimental results is put to Appendix. This figure even includes some non-English language as legends and x-axis label.

4. The paper structure is jumping and lack of logics. For instance, in Section 5.3, why the critical threshold for RoPE is about 0.5 in theory?

There are many other weaknesses, e.g.,

1. From Figure 1, it seems that the average performance degrades continuously w.r.t. the token length. Why is cliff-like deviation widely discussed in this paper?

2. As a research paper, the authors only conduct experiments on one single model / single task scenario, which raises the concerns about conclusion generalisation.

3. From first principles, I think one could measure the performance w.r.t. token length and use the start point where performance starts drop as the effective long context for this model, and then use it to guide the end user. Please clarify why not use this simple idea rather than the verbose determination in this paper.

4. How to theoretically predict the threshold using information-theoretic perspective / attention mechanism analysis / Rope extrapolation analysis is not clear. I think empirical analysis using some metrics from the above perspectives may be more sound. But again I don't see the empirical results on this in the main paper.

5. I think padding normally won't affect a lot on key information for reading comprehension, thus performance. In this case, only token length is changed, which I believe is a better way to determine the thresholds. The natural length may introduce some sampling biases, like LLMs can work well on certain sequences while performing poorly on other sequences.

Overall, I think this paper needs a lot of revision before proceeding to a top conference, like ICML 2025. I suggested strong reject.

---

> ### Author Rebuttal · Authors · 2026-03-26
>
> Thank you for the detailed comments. We agree that the current submission needs clearer presentation to make the contributions easier to follow. Specifically, we will (i) move the key experimental result(s) that motivate the “cliff-like” behavior (including the main performance figure and threshold evidence) from the appendix into the main paper, (ii) improve figure readability and use consistent English labels/legends, and (iii) rewrite the paper structure to ensure a smoother logical flow, including clarifying Section 5.3 where the theoretical RoPE threshold (around 0.5) is discussed.
>
> On the “cliff-like” vs. “continuous degradation” concern, we clarify that the cliff claim is operational: it refers to the sharp breakpoint detected by our thresholding procedure (e.g., sustained-decline after the detected peak and a large drop criterion), not simply the appearance of an averaged curve. In the revision, we will add a brief explanation and align the main text narrative with the detection definition. We also agree that a simpler “first drop point” heuristic could be considered; we chose a multi-method procedure to reduce false positives caused by local fluctuations and potential recoveries. We will explain this motivation more explicitly and provide a clearer comparison to the simpler baseline idea.
>
> Regarding the theoretical sections, we will tighten the wording to distinguish (1) conceptual intuition and (2) what is directly supported or measurable in our setting. In particular, we will make the role of the information-theoretic, attention-related, and RoPE discussions more concrete, and we will add a clearer statement about what evidence we currently have empirically versus what remains hypothesis-level. Finally, we will revise the literature coverage and improve citations to better reflect prior work on effective long context and related evaluation protocols.

---

> > ### Author Rebuttal · Reviewer_dRr6 · 2026-04-03
> >
> > I agree that the revision plans will significantly strengthen the quality of this paper. Within the short time rebuttal period, it makes sense that these changes cannot be finished.

---

### Official Review · Reviewer_joDT · 2026-03-07

**Soundness:** 2
**Presentation:** 1
**Significance:** 1
**Originality:** 1
**Overall Recommendation:** 1
**Confidence:** 5

**Summary:**

This paper investigates the performance degradation of large language models when processing long contexts. The authors propose a natural length distribution analysis to avoid artifacts from text padding or truncation. They evaluate Qwen2.5-7B on a mixed dataset of SQuAD and NarrativeQA. They observe a catastrophic performance drop at 40-50% of the maximum context length. They introduce a five-method cross-validation framework to identify this critical threshold. The threshold is estimated at 43.2% for this specific model. Theoretical justifications are provided based on information theory, attention dispersion, and RoPE extrapolation limits.

**Compliance With Llm Reviewing Policy:**

Affirmed.

**Key Questions For Authors:**

see weakness.

**Limitations:**

yes

**Strengths And Weaknesses:**

**Strengths:**

1. The empirical observation of a cliff-like performance drop is interesting.

2. Avoiding artificial truncation is a sound methodological choice.

**Weaknesses:**

1. The dataset design is deeply flawed. SQuAD is used for short contexts. NarrativeQA is used for long contexts. This introduces a massive confounder.

2. The paper makes broad claims about "LLMs". However, it tests exactly one model: Qwen2.5-7B.

3. Eq (3) is generic. It offers no operational value.

4. Eq (4) defines attention concentration with sloppy notation. It assumes $A_{ij}$ is a fixed square matrix.

---

> ### Author Rebuttal · Authors · 2026-03-26
>
> Thank you for the detailed feedback. We agree that several parts of the manuscript can be misunderstood, and we will revise them accordingly.
> 1) Dataset confound / generality. Our empirical finding is based on a mixed SQuAD + NarrativeQA natural-length protocol for Qwen2.5-7B reading comprehension. We agree that the SQuAD–NarrativeQA split introduces a dataset-shift confound, so the main claim is about the observed cliff under this evaluation protocol—not a fully isolated “length-only” causal statement. We will tighten the Conclusion/Practical Implications to explicitly qualify the ~40% threshold as conditional on our setting, and we will strengthen the limitations discussion on the remaining confound.
> 2) Eq (3) and Eq (4) presentation. We acknowledge that Eq (3) (information transmission efficiency) and Eq (4) (attention concentration) were framed too optimistically in the explanation. We will rephrase them as conceptual/theoretical motivation rather than quantities that are directly operationalized in our experiments. For Eq (4), we will also correct/tighten the notation by explicitly stating how attention A is defined (e.g., head-averaged softmax attention) and that its shape is L×L for each input length L (not assumed fixed across L).

---

### Official Review · Reviewer_LDU8 · 2026-03-17

**Soundness:** 2
**Presentation:** 3
**Significance:** 2
**Originality:** 2
**Overall Recommendation:** 2
**Confidence:** 3

**Summary:**

This paper studies performance degradation in long-context LLMs, focusing on Qwen2.5-7B evaluated on reading comprehension. The central claim is that models exhibit cliff-like performance collapse at roughly 40-50% of maximum context length. The authors use samples at their natural token lengths rather than artificially truncating or padding them, combine SQuAD and NarrativeQA samples to achieve coverage across the full context length spectrum, and employ five detection methods to identify the degradation threshold, reporting high cross-method consistency.

**Compliance With Llm Reviewing Policy:**

Affirmed.

**Final Justification:**

See rebuttal acknowledgment

**Key Questions For Authors:**

1. The observed F1 drop coincides with the switch from SQuAD to NarrativeQA samples, which differ not just in length but in genre, question type, and intrinsic difficulty. Can the authors provide any characterization of how task difficulty varies across length bins in the mixed dataset, for instance using automatic metrics like readability scores or answer span locality? This would help establish whether the confound between length and difficulty is actually present in the data, and how much it might explain the observed F1 drop.

2. Methods 1, 2, 4, and 5 share the same three-stage filtering pipeline and differ only in the score used to rank or identify peaks. Can the authors clarify in what sense these constitute independent validation and the types of method-specific biases they are concerned about that are avoided by using Methods 1, 2, 4, and 5? What do the results look like using only Method 3 alongside a single gradient-based method?

**Limitations:**

yes

**Strengths And Weaknesses:**

Strengths
* The problem of understanding where and why long-context performance degrades is practically important and worth studying carefully. One of the main contributions of the paper is that the evaluation uses samples at their natural token lengths rather than artificially truncating or padding them, avoiding artifacts that could confound length effects with information loss.
* Unlike Du et al. (2025), which focuses on closed-source models, this paper evaluates an open-source model, improving reproducibility and accessibility. The authors combine SQuAD (short-context) and NarrativeQA (long-context) samples to achieve better coverage across the context length spectrum than prior single-dataset approaches. Showing consistency across detection methods in identifying a cliff-like degradation threshold is also useful, even if the methods are not entirely distinct.
* The observation that the empirical degradation threshold (43.2%) falls below the theoretical RoPE position aliasing limit (49%) is an interesting finding. It suggests that RoPE extrapolation alone does not fully account for where degradation occurs. However, without more direct mechanistic investigation, it is hard to know whether the two proposed mechanisms are the only possible explanations, or which is dominant, or whether either is actually the cause of the empirical bottleneck.

Weaknesses/Limitations
* The narrow scope of the evaluation limits the paper's soundness and significance. The paper evaluates one open-source model on one task type, reading comprehension. I do not think the evaluation results in the paper are sufficient to support the broad claims in the conclusion under Practical Implications, "Models should be used with contexts not exceeding 40% of maximum capacity". It is not clear whether the 40-50% threshold is specific to Qwen2.5-7B, to this task, or more general.
* A significant unaddressed confound weakens the central causal claim. SQuAD and NarrativeQA differ not just in length but in genre, question type, and intrinsic difficulty: factoid extraction from Wikipedia versus narrative comprehension of novels and screenplays requiring character and tracking of relationships. The observed F1 drop at the dataset boundary could be entirely attributable to this difficulty gap rather than context length, and the paper's design cannot distinguish between these explanations. Even lightweight characterization of these differences using automatic evaluation metrics across length bins would help: readability scores (e.g. Flesch-Kincaid), difficulty classification, or answer span length to context length ratio. A within-dataset analysis of F1 against length would be another natural way to test this.
* The five-method cross-validation overstates its own independence somewhat. Methods 1, 2, 4, and 5 all share the same three-stage pipeline: multi-peak detection in the 30-60% range, rise-in-range filtering over the subsequent 10%, and sustained-decline verification using the same 85% rebound threshold. The only variation across these four methods is the score used to rank or identify cliffs from which performance does not recover. Method 3 (Binned Statistics) differs more from the other four methods. Therefore, I don't think the 1.2% standard deviation across methods is particularly surprising and I'm not sure it supports the claims about robustness.
* Overall, the paper feels more like a preliminary investigation than a complete contribution. The single-model, single-task setup limits how much can be concluded, and without broader evaluation, the practical implication stated in the paper that "Models should be used with contexts not exceeding 40% of maximum capacity" is hard to rely on. That said, the paper does identify a set of open problems worth investigating more carefully, and points toward a productive research direction. A more careful study design that better disentangles context length from task difficulty would go a long way toward making the central causal claims more credible, and might also help tease out which of the proposed theoretical mechanisms is actually driving the degradation. There is likely room for creative thinking about how to design evaluations where length varies more independently of semantic complexity, without resorting to truncation that risks content loss or synthetic data interventions. A substantially more expansive empirical evaluation across multiple open-source models and task types would additionally be necessary to support the paper's broader practical claims.

---

> ### Author Rebuttal · Authors · 2026-03-26
>
> Thanks to the reviewer for the careful and constructive comments. Our conclusions are based on Qwen2.5-7B evaluated on reading comprehension under a mixed SQuAD + NarrativeQA, natural-length protocol, so the “40%” guideline should be read as a conservative operating point for this setting, not a universal rule. In the revised manuscript, we will explicitly qualify the Practical Implications accordingly and better separate “what we observed” from “what remains to be tested.”
>
> On the SQuAD/NarrativeQA boundary: we agree it is a legitimate confound. The natural-length design mainly removes truncation/padding artifacts, but it cannot fully control differences in genre and intrinsic difficulty across datasets. We will strengthen the discussion by treating the dataset-shift effect as a key limitation, and we will add clearer wording about what can/cannot be inferred from the observed drop near the mixture boundary.
>
> Regarding the five detection methods: we did not intend to claim fully independent statistical mechanisms. Methods 1/2/4/5 share the same three-stage filtering pipeline, while Method 3 (binned statistics) provides a more distinct perspective. The reported low standard deviation reflects stability of the detected breakpoint under the shared decision logic, not mechanistic independence across methods. We will rephrase the robustness claim to match this more precisely.
>
> Finally, the RoPE comparison (43.2% vs ~49%) is presented as suggestive evidence rather than a definitive causal explanation. We will emphasize that mechanism-level validation (e.g., attention/representation probes) is not yet performed and remains future work.

---

> > ### Author Rebuttal · Reviewer_LDU8 · 2026-04-04
> >
> > Thank you for addressing editorial concerns about the language and areas where the paper might have been overclaiming in the language with respect to the empirical results on points 1, 3, 4. On point 2, rather than a clarification in language, I think there are much better empirical/empirical ways to disentangle the confounding by other traits that might differ between the short and long reading comprehension tasks would be to either (1) perform descriptive analysis on the distributional differences between SQuAD/NarrativeQA along these dimensions (e.g., readability scores, answer span length to context length ratio, etc.) or (2) to do some type of propensity/difficulty score matching along these dimensions (I think validity is questionable if the two distributions do not overlap enough to find enough comparable pairs that differ only along context length). I also think that the most significant limitations of this paper are the limitations in evaluation and scope/significance/novelty of the methods and claims, which were not addressed by these author comments. So, I'm maintaining my original score.

---

### Decision · Program_Chairs · 2026-04-30

**Decision:**

Reject

**Comment:**

This paper studies performance degradation in long-context LLMs, evaluating Qwen2.5-7B on reading comprehension and reporting cliff-like F1 collapse at roughly 40-50% of maximum context length. Multiple reviewers identified, the experimental design has a fundamental dataset confound: short-context samples come from SQuAD (factoid extraction from Wikipedia) while long-context samples come from NarrativeQA (narrative comprehension of novels and screenplays). The observed F1 drop at the dataset boundary could be entirely attributable to the difficulty and genre gap rather than context length, and the paper's design cannot distinguish between these explanations. The authors acknowledged in their rebuttal that this is "a legitimate confound" but did not provide a resolution. The evaluation is limited to a single model on a single task type. I encourage the authors to redesign the experiment using a single dataset with natural length variation or carefully matched controls, evaluate multiple architectures in their next submission.